# Effects of Substrates on Nucleation, Growth and Electrical Property of Vertical Few-Layer Graphene

**DOI:** 10.3390/nano12060971

**Published:** 2022-03-15

**Authors:** Tianzeng Hong, Chan Guo, Yu Zhang, Runze Zhan, Peng Zhao, Baohong Li, Shaozhi Deng

**Affiliations:** State Key Laboratory of Optoelectronic Materials and Technologies, Guangdong Province Key Laboratory of Display Material and Technology, School of Electronics and Information Technology, Sun Yat-sen University, Guangzhou 510275, China; hongtz@mail2.sysu.edu.cn (T.H.); guoch5@mail2.sysu.edu.cn (C.G.); zhanrz3@mail.sysu.edu.cn (R.Z.); zhaop36@mail2.sysu.edu.cn (P.Z.); lbaoh@mail.sysu.edu.cn (B.L.)

**Keywords:** vertical few-layer graphene, nucleation, substrates, field electron emission

## Abstract

A key common problem for vertical few-layer graphene (VFLG) applications in electronic devices is the solution to grow on substrates. In this study, four kinds of substrates (silicon, stainless-steel, quartz and carbon-cloth) were examined to understand the mechanism of the nucleation and growth of VFLG by using the inductively-coupled plasma-enhanced chemical vapor deposition (ICPCVD) method. The theoretical and experimental results show that the initial nucleation of VFLG was influenced by the properties of the substrates. Surface energy and catalysis of substrates had a significant effect on controlling nucleation density and nucleation rate of VFLG at the initial growth stage. The quality of the VFLG sheet rarely had a relationship with this kind of substrate and was prone to being influenced by growth conditions. The characterization of conductivity and field emissions for a single VFLG were examined in order to understand the influence of substrates on the electrical property. The results showed that there was little difference in the conductivity of the VFLG sheet grown on the four substrates, while the interfacial contact resistance of VFLG on the four substrates showed a tremendous difference due to the different properties of said substrates. Therefore, the field emission characterization of the VFLG sheet grown on stainless-steel substrate was the best, with the maximum emission current of 35 µA at a 160 V/μm electrostatic field. This finding highlights the controllable interface of between VFLG and substrates as an important issue for electrical application.

## 1. Introduction

Substrates are indispensable carriers for electronic devices using nano-materials which have significant influence on its performance, especially for field emission cold-cathode devices. Among nano-materials, vertical few-layer graphene (VFLG) has been considered one of the most ideal field emission materials due to its excellent electrical properties and sharp edges, etc. [1,2,3,4,5,6]. Tang et al. reported that the field emission current density of VFLG grown on a tungsten (W) tip reached 5.85 × 10^8^ A/cm^−2^ [7]. Jiang et al. reported that VFLG grown on a copper (Cu) substrate had far better field emission characteristics, which had a turn-on electric field of 1.3 V/μm, a threshold field of 3.0 V μm^−1^ and field-enhancement factor of 1.1 × 10^4^ [8]. Wang et al. found that VFLG on a silicon (Si) substrate with a different surface topography results a different shape and further influences its field emission characteristics [9]. Apparently, VFLG shows different field emission characteristics when grown on various substrates. Understanding the relationship of substrates and the electrical properties of VFLG is important for electronic devices application of VFLG. 

VFLG can be prepared on different substrates with or without catalysts at a low temperature through plasma-enhanced chemical vapor deposition (PECVD) [10,11,12,13,14]. The impact factors of VFLG growth, such as plasma power, gas ratio, pressure, growth time, temperature and so on, have been studied in detail for the controllable growth of VFLG regarding growth rate, morphology and density of VFLG [15,16,17,18]. Researchers have also noticed that the nucleation, density, defects and single crystals of VFLG grown on different substrates are related to the substrate material [19,20,21]. For example, Ghosh et al. reported that the morphology, growth rate and film quality of VFLG are influenced by substrates which determine the deposition of nanographitic layer and guide the growth of VFLG [21]. More studies are needed to clarify the detailed influence factors of substrates for the nucleation and growth of VFLG, especially in regard to the electrical characteristics.

The aim of this paper was to study the growth process of VFLG on different substrates and to explore the relationship of substrates and the electrical properties of VFLG. The initial nucleation and growth stage of VFLG on Si, stainless-steel, quartz and flexible carbon-cloth substrates were studied in detail. Crystallinity and microstructure were characterized by high resolution transmission electron microscopy (HRTEM). Conductivity and field emission measurements were carried out to understand the relationship of substrates on the electrical properties of VFLG.

## 2. Materials and Methods

### 2.1. Preparation of VFLG

The VFLG was synthesized using a one-step method with inductively-coupled plasma-enhanced chemical vapor deposition (ICPCVD) [22]. The substrates of Si wafer, stainless-steel, flexible carbon-cloth and quartz were placed in an ICPCVD reactor simultaneously. The chamber was pumped to 5 × 10^−4^ Torr by a mechanical pump and a molecular pump. Meanwhile the substrates were heated to 800 °C. Then the substrates were pretreated by applying a radio frequency power of 900 W at an Ar (15 sccm) and a H_2_ (15 sccm) atmosphere. A negative bias voltage (100 V) was applied to enhance the plasma energy on the substrates. After 15 min, a mixture of H_2_ (10 sccm) and CH_4_ (60 sccm) was fed into the ICPCVD chamber as the hydrocarbon source for VFLG growth. The radio frequency power increased to 1100 W with the negative bias voltage of 100 V. After the reaction, the gas and power were turned off, the chamber was quickly cooled to room temperature and the samples were removed. 

### 2.2. Characterization 

The morphology, structure, composition and crystallinity of VFLG were characterized by Scanning Electron Microscopy (SEM) (Supra 60, Zeiss, Oberkochen, Germany), HRTEM (Titan3 G2 60-300, FEI, Hillsboro, OR, USA), Raman Spectroscopy (In Via Reflex, Renishaw, Wotton-under-Edge, Gloucestershire, UK) with a 532 nm laser and a water contact angle (SDC280E, SINDIN, Dongguan, China), respectively. The conductivity and field emission characteristics of individual VFLG were measured using a nano-probe measurement method, in which a manipulator with tungsten nano-probes in SEM system was adopted.

## 3. Results and Discussion

To elucidate the influence of substrates on the nucleation and growth of VFLG the VFLG were grown on semiconductor Si, metal stainless-steel, flexible carbon-cloth and insulator quartz substrates through ICPCVD under the same growth conditions. Figure 1 shows SEM images of VFLG grown for 20 min on the above substrates. This VFLG has a higher growth density and larger height on stainless-steel and quartz substrates compared to that grown on Si and flexible carbon-cloth substrates. In a previously published study, it was reported that the growth density and growth rate could also be affected by the growth time, RF power and radicals concentration, etc. [23]. However, in this experiment all the parameters were kept the same, thus the differences were attributed to the substrate.

On the other hand, the morphology of VFLG grown on the above four substrates was the same as shown in Figure 1e–h. All four VFLGs showed a petal-like shape which grew anisotropically in both vertical and horizontal directions from the nucleation point. After nucleation, the growth of VFLG increased beyond the substrate; thus the substrate had little influence on the morphology of the VFLG. The morphology of VFLG was mainly influenced by the plasma density, RF power, gas rate, and temperature, etc. [24,25].

The adhesion of the VFLG on the four substrates was tested through high pressure gas blowing (N_2_ at 0.1 MPa). The VFLGs remained attached on the Si, stainless-steel and quartz substrates, while they peeled off completely from the flexible carbon-cloth substrate, as shown in the Appendix A. The low adhesive power of VFLG on the flexible carbon-cloth substrate can be ascribed to low surface energy, which will be discussed in detail below.

The initial nucleated stages of the VFLG were studied in order to understand the effects of the substrates on the nucleation process of VFLG. According to the thermodynamic theory of nucleation [26], the rate of nucleation (*N_i*_*) can be described as below:(1)$$N_{i^{*}}={Ra_{0}}^{2}n_{0}{\left(\frac{R}{n_{0}v}\right)}^{i^{*}}\exp(\frac{(i^{*}+1)E_{des}-E_{s}-E_{i^{*}}}{k_{B}T})\;\;\;\;\;\;\;\;\;\;({\mathrm{cm}}^{-2}{\;s}^{-1})\;$$
where *R* is the deposition rate; *a*_0_ is the lattice constant of graphene; *n*_0_ is the number of nucleation sites; *v* is the frequency of atom vibration (~10^13^ s^−1^); *i** is the atom number of nucleation; *k_B_* is the Boltzmann constant; *T* is the temperature; and *E_des_*, *E*_*s*_ and *E*_*i**_ are desorption activation energy, surface diffusion activation energy and nucleation energy, respectively. To achieve a high nucleation rate, the *E_des_* should be as high as possible.

The *E_des_* also increased with the increasing surface energy [27]. In this experiment, the surface energy of substrates was characterized by water contact angle analysis (Appendix A). Higher surface energy exhibited a smaller contact angle. The contact angles of Si, flexible carbon-cloth and quartz substrates were 66°, 134° and 50°, respectively (Appendix A). After the plasma bombardment pretreatment the contact angles of the Si, flexible carbon-cloth and quartz substrates decreased to 32°, 116° and 35°, respectively, which indicated that pretreated substrates have higher surface energies compared to non-pretreated substrates (Appendix A). The stainless-steel substrate is not discussed here due to its catalyst properties. The catalyst property of the substrate was able to overcome the influence of surface energy. Then all the substrates were used for a 5 min VFLG growth. No VFLGs grew on the non-pretreated substrates as shown in Figure 2a–c, while VFLGs grew on the pretreated substrates. Among the pretreated substrates, VFLGs on the Si and quartz substrates had higher nucleation density and growth rate than those on carbon-cloth, as shown in Figure 2d–f. Obviously, the substrates with high surface energy resulted in high nucleation density and growth rate. The high surface energy of substrates had high *E_des_* which was heavily absorbed by the carbon radicals on the substrate, allowing them to stay attached for a long period of time to form stable nucleation. The substrate’s physical properties and surface microstructure were able to regulate its surface energy, then influence VFLG nucleation.

In addition, low surface energy and *E_des_* also led to the low adhesive power of VFLG on substrate. Low *E_des_* led to short retention times and the easy desorption of carbon radicals on the substrates during the nucleation process [21,28]. Neither time nor carbon radicals were sufficient for the formation of high crystallinity or a large sized nucleation interface on the substrate. This is the reason for the bad adhesion of VFLG grown on flexible carbon-cloth substrates.

The different catalyses of substrates also led to different nucleation rates. Stainless-steel, Si and carbon-cloth substrates were taken as examples, as shown in Figure 3. Because the stainless-steel substrate contains Ni, Fe catalyst elements, the VFLG had a fast nucleation rate in 2 min (Figure 3a) and growth larger in 5 min (Figure 3b). In comparison, Si and flexible carbon-cloth substrates, which had no catalysis, underwent no nucleation of VFLG in a 2 min period of growth (Figure 3c,e). To speed up the nucleation rate, a 20 nm Fe catalyst film was sputtered on the Si and flexible carbon-cloth substrates. Then, in a 2 min period of VFLG growth, the nucleation rate and growth rate of VFLG increased obviously (Figure 3d,f). Catalysts were able to decrease *E*_i*_ to nucleate at lower energies and increase reaction rates to accelerate the nucleation and growth rate of VFLG at the initial growth stage.

Based on the above experimental results, surface energy and the catalysis of substrates have significant influence on the nucleation density and growth rate of VFLG, and also have a synergistic effect on the nucleation and growth of VFLG (Table 1). Under the same growth conditions, substrates with high surface energy and catalysts were able to promote the nucleation and growth of VFLG. 

The quality of VFLG film on different substrates was studied using Raman spectroscopy. Figure 4 shows the Raman spectra of the VFLG film on the Si, stainless-steel, flexible carbon-cloth and quartz substrates. The Raman spectrum of the VFLG shows typical graphene peaks: D peak at ~1350 cm^−1^, G peak at ~1580 cm^−1^ and 2D peak at ~2700 cm^−1^, respectively. The high intensity of the D peak indicates that a large number of defects and edges exists in the VFLG film. *I_D_/I_G_* of the VFLG film on the stainless-steel and quartz substrates was higher than 1 which can be attributed to the high-density sharp edges of VFLG. The presence of defects and sharp edges can introduce the D’ peak at 1620 cm^−1,^ leading to the broadening of the G peak (Table 2). The broad 2D peaks and *I_2D_/I_G_* less than 1 indicate the VFLG grown on the four substrates were few-layer graphene. The results indicate VFLGs have similar qualities on different substrates. Although Raman is an efficient way to characterize the quality of graphene, it cannot accurately show the structure of a single VFLG sheet.

To further determine the crystallinity and surface morphology of the VFLG, HRTEM was adopted. Figure 5 shows the top morphology of the VFLG grown on different substrates. Disordered and amorphous carbon structures exist and introduce defects onto the surface of the VFLG (the red circle in Figure 5a–d). Hexagonal carbon structures can be seen clearly in the HRTEM images of the VFLG, which confirms the typical signature of graphene (Figure 5e–h). The middle and bottom of the VFLG both have fine structures and are thicker than the top. Disordered structures are shown on the concentric ring diffraction pattern of the VFLG (Appendix A). All in all, the crystallinity of the VFLGs on different substrates was almost the same and was rarely influenced by substrates under the same growth conditions. TEM and Raman results demonstrate that the final quality of the VFLG after full growth was the same, which proves that the substrate only affects the nucleation stage and not the growth stage.

The electrical conductivity properties of VFLG on different substrates were characterized using a nano-probe measurement system. As shown in Figure 6a, a single VFLG sheet was cut off the substrate and two tungsten nano-probes were brought into contact with the top and bottom of the VFLG to measure its body resistance. The results show that the VFLG which had grown on the four substrates all showed similar *I*-*V* characteristics (Figure 6b). The resistance of a single VFLG sheet grown on Si, stainless-steel, flexible carbon-cloth and quartz substrates was 7.8 × 10^3^ Ω, 3.1 × 10^3^ Ω, 4.7 × 10^3^ Ω and 2.1 × 10^3^ Ω, respectively. Our results were similar to the reported sheet resistance of pristine graphene (0.6–1.8 KΩ/sq) [29,30,31]. Even though the substrates determined the nucleation of the VFLG, it rarely interfered with the growth process of VFLG after nucleation. The same structure of the VFLG had the similar resistance. Therefore, the body resistance of the VFLG was independent of the substrates. Xu et al. also reported that the substrate did not alter the intrinsic optical conductivity of a 2D crystal [32]. 

In contrast, the electrical conductivity properties of the VFLG on the substrate showed differences. In this measurement, one probe was brought into contact with the top of the VFLG and the other probe with the substrate. The total resistance of the VFLG and that of the substrate were measured together (including the VFLG body resistance, the contact resistance and the substrate resistance). Figure 6c–e shows the *I*-*V* characteristics of the VFLG on Si, stainless-steel and flexible carbon-cloth substrates. According to nonlinear rectifying behavior of the Schottky contact, it is obvious that the interface between the VFLG and the Si substrate was not Ohmic contact but Schottky contact (Figure 6c). Compared to the Si substrate, the *I-V* characteristics of the VFLG on stainless-steel and flexible carbon-cloth show good Ohmic contact (Figure 6d,e). Total resistance of the VFLG on Si, stainless-steel, flexible carbon-cloth substrates was 2.4 × 10^4^ Ω, 3.3 × 10^4^ Ω, 2.07 × 10^4^ Ω, respectively. The resistance of probe to probe, probe to Si substrate, probe to stainless-steel substrate and probe to flexible carbon-cloth substrate was about 4.76 Ω, 2127 Ω, 4.96 Ω and 96.15 Ω, respectively (Appendix A), which means the substrate resistance was small and, other than that of the Si substrate, can be ignored. Therefore, the contact resistance at the interface is the main type that is an order of magnitude larger than the VFLG body resistance. The contact interface of the VFLG and the substrates play a key role in determining the resistance of the VFLG on substrates and should be reduced when aiming for device application.

The field electron emission characterization of the VFLG on Si, stainless-steel and flexible carbon-cloth substrates were measured to further explore the influence of substrates on VFLG. VFLG on quartz is not measurable because of the high resistivity of the substrate. Field emission characterization of a single VFLG sheet was characterized using the same nano-probe measurement system. The voltage was applied to anode, specifically the tungsten probe. The substrate was set to be a cathode emitter. The distance of the VFLG and the probe was 1 μm. The inset of Figure 7 shows the circuit diagram of the VFLG field electron emission. It is obvious that the field emission characterization of the VFLG was influenced by the substrate. The turn-on field of the VFLG on the Si, stainless-steel and flexible carbon-cloth substrates are 158 ± 30 V/μm, 118 ± 16 V/μm and 114 ± 16 V/μm, respectively, at a current of 1 nA. At the start of the field emission, temperature and adhesive power did not influence it. The turn-on field of the VFLG on stainless-steel and flexible carbon-cloth substrate was similar. The Schottky contact of the VFLG on the Si substrate led to a high contact resistance, resulting in a high turn-on field. The VFLG grown on the flexible carbon-cloth substrate showed the weakest emission current, which is only 0.5 μA under the breakdown voltage; correspondingly, the current density was 1.4 × 10^4^ A/cm^2^ (Figure 7e). The main reason for this can be ascribed to the low adhesive power of the VFLG on flexible carbon-cloth, where the VFLG easily detached from the flexible carbon-cloth substrate, causing vacuum breakdown. The VFLG grown on the Si and stainless-steel substrates presented better field emission currents. The maximum emission current of the VFLG grown on the Si substrate reached 10 μA at 250 V/μm before vacuum breakdown; correspondingly, the current density was 2.2 × 10^5^ A/cm^2^ (Figure 7a). The maximum emission current of the VFLG grown on the stainless-steel substrate reached 35 μA at 160 V/μm before vacuum breakdown; correspondingly, the current density was 7.8 × 10^5^ A/cm^2^ (Figure 7c). The difference between the two was attributed to the electrical and thermal conductivity of substrates. The electrical conductivity of n-Si and stainless-steel was 1.56 × 10^−3^ S/m and 9 × 10^6^ S/m and the thermal conductivity of n-Si and stainless-steel was 133 W/m K and 14 W/m K (Appendix A). High thermal conductivity can help heat dissipation and is beneficial for the VFLG sustaining a high current during the field electron emission process [7]. Excellent electrical conductivity and thermal conductivity strongly promotes the field emission characterization of the VFLG. The corresponding *F-N* plot is linear in the low field regions and fit with the field emission *F-N* theory (Figure 7b,d,f). The field enhancement factor of VFLG can be calculated according to the *F-N* theory, which is applied to the classical field emission process and described as: (2)$$J=\frac{A{\left(\beta E\right)}^{2}}{\varphi }\exp\left(-\frac{B\varphi^{3/2}}{\beta E}\right)$$
where *A* = 1.541434 × 10^−6^ A eV V^−2^, *B* = 6.830890 eV V nm^−1^, *β* field enhancement factor, *φ* work function of sample, *E* electrical field.

The work function of VFLG can be assumed to be constant at 4.5 eV [33]. The field enhancement factor of the VFLG on the Si, stainless-steel and flexible carbon-cloth substrates is 21 ± 2.4, 33 ± 9.1 and 26 ± 5.7, respectively. The small field enhancement factor can be ascribed to the short distance of the probe to the VFLG [34]. Similar field enhancement factors of VFLG further indicate the important of substrates for the field emission of VFLG. The results show that the adhesive power and electrical properties of substrates are crucial for the field emission characteristics of VFLG.

Finally, the maximum field emission current of the VFLG is compared with some single sheet/wire/tube nanomaterials with excellent field emission performance in Table 3. The VFLG performed better than the CuO nanoneedle, SiC nanowire and Mo nanoscrew but not good as the LaB_6_ nanowire, carbon nanotube and multilayer graphene. Current density of VFLG is larger than the SiC nanowire and Mo nanoscrew, but lower than the LaB_6_ nanowire and multilayer graphene. The breakdown voltage of VFLG is lower than that of SiC nanowire and LaB_6_ nanowire. The limitations of the VFLG field emission characteristics can be ascribed to low adhesive power and low crystallinity. Improving the quality of VFLG is crucial for better field emission characteristics. 

## 4. Conclusions

Here, the mechanism of the nucleation and growth of VFLG on different substrates was studied. The high surface energy of substrates can enhance *E_des_*, which is good for the adsorption of carbon radicals and the nucleation of VFLG. Catalysts decrease *E_i_*_*_ and increase reaction rate, leading to a high nucleation rate. The above factors have significant effects on controlling the nucleation density and growth rate of VFLG during the initial growth stage. The quality of VFLG rarely has a relationship with this kind of substrate and is prone to being influenced by growth conditions, such as RF power, temperature, gas ratio, etc. Nucleation of VFLG determines the contact interface of VFLG on substrate, further influencing the adhesive power of VFLG. Poor adhesive power restrains the field emission and electrical conductivity properties of VFLG. The resistance of a single VFLG sheet is rarely influenced by the substrates. Contact interface of the VFLG on substrates is a key factor in determining the resistance of the VFLG on substrates. The VFLG grown on Si and stainless-steel substrates has good field emission characteristics compared with the VFLG grown on the flexible carbon-cloth substrate due to better adhesive power. These findings suggest that the substrate has an influence on the formation of VFLG. The controllable interface of between VFLG and the substrate is important for electrical and field emission applications. 

## Figures and Tables

**Figure 1 nanomaterials-12-00971-f001:**
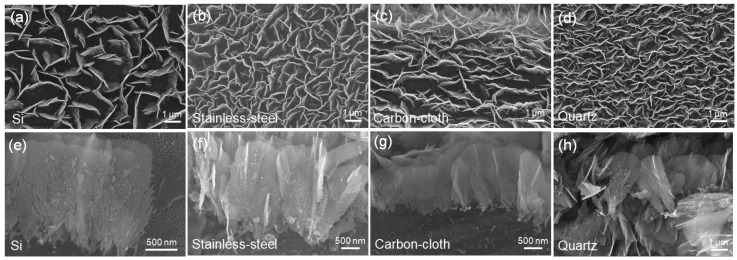
Top-view and side-view of the morphology of VFLG on different substrates. (**a**,**e**): Si substrate; (**b**,**f**): stainless-steel substrate; (**c**,**g**): flexible carbon-cloth substrate; (**d**,**h**): quartz substrate.

**Figure 2 nanomaterials-12-00971-f002:**
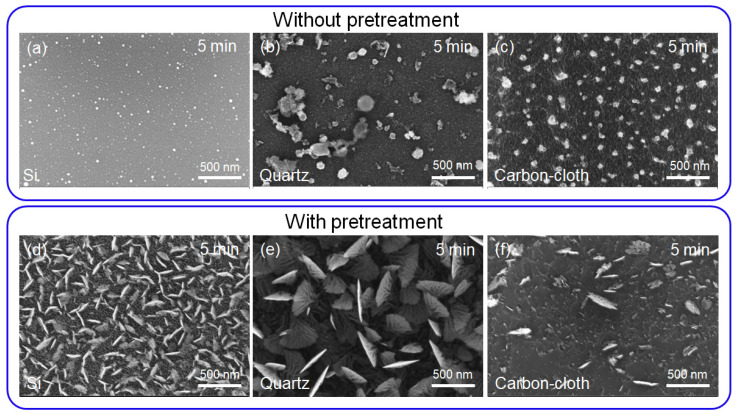
The surface energy of substrates’ influence on VFLG growth. (**a**–**c**) represent a VFLG growth time of 5 min on different substrates without pretreatment, where (**a**) is Si, (**b**) is quartz and (**c**) is flexible carbon-cloth. (**d**–**f**) show a VFLG growth time of 5 min on different substrates with pretreatment, where (**d**) is Si, (**e**) is quartz and (**f**) is flexible carbon-cloth.

**Figure 3 nanomaterials-12-00971-f003:**
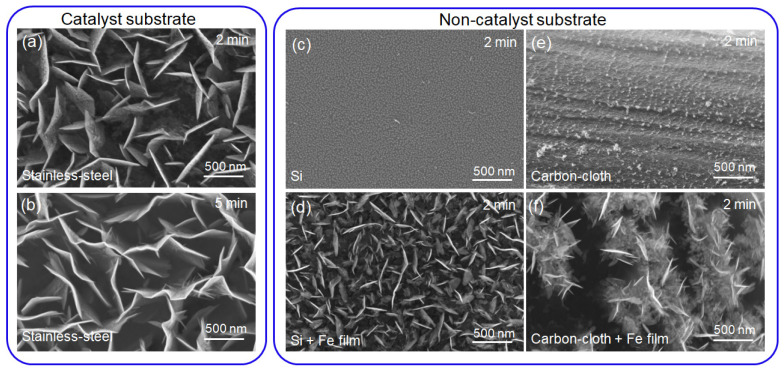
The influence of catalysis on VFLG growth. VFLG growth of 2 min (**a**) and 5 min (**b**) on stainless-steel substrates with pretreatment. VFLG growth of 2 min on (**c**) Si, (**d**) Si + Fe film, (**e**) flexible carbon-cloth and (**f**) flexible carbon-cloth + Fe film substrates with pretreatment.

**Figure 4 nanomaterials-12-00971-f004:**
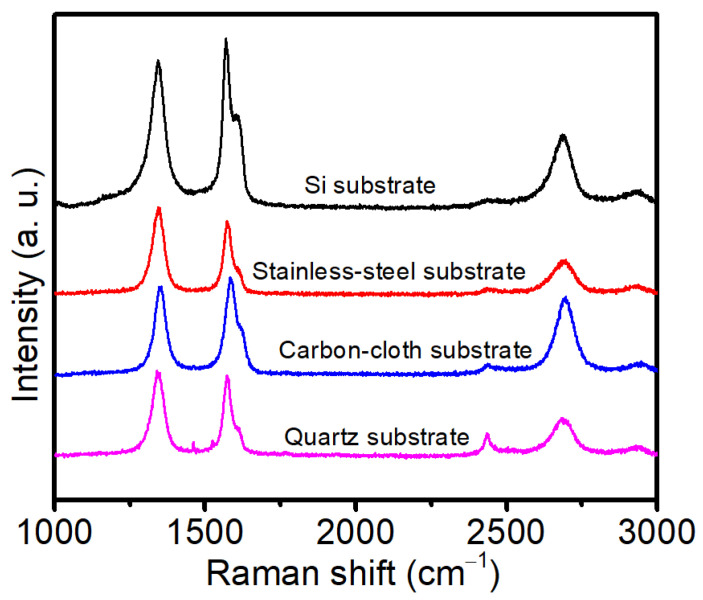
The Raman spectra of VFLG grown on different substrates.

**Figure 5 nanomaterials-12-00971-f005:**
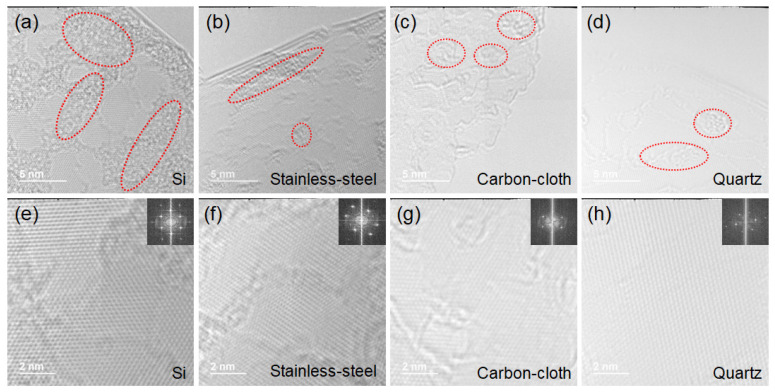
HRTEM images of the top area of the VFLG grown on Si, stainless-steel, flexible carbon-cloth and quartz substrates. The insets of (**e**–**h**) are the diffraction patterns of the VFLG. Red circles of (**a**–**d**) show the typical amorphous carbon structure areas on the VFLG sheet.

**Figure 6 nanomaterials-12-00971-f006:**
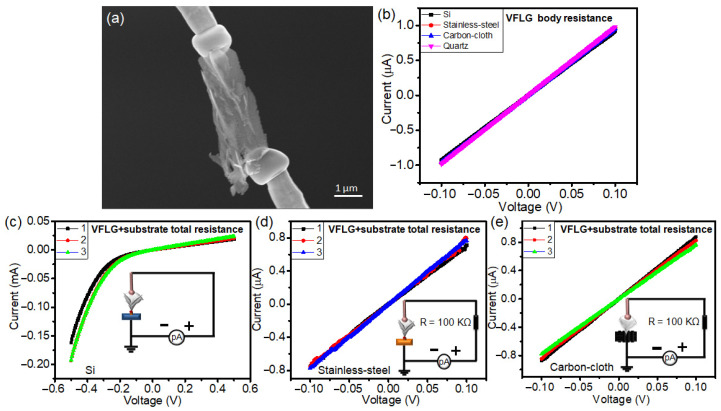
(**a**) SEM image of a double probe electrical test of a single VFLG sheet cut off the substrate. (**b**) *I-V* curves of the single VFLG sheet cut off stainless-steel, Si, flexible carbon-cloth and quartz substrates. (**c**) *I-V* curves of three randomly selected single VFLGs on a Si substrate. (**d**) *I-V* curves of three randomly selected single VFLGs on a stainless-steel substrate. (**e**) *I-V* curves of three randomly selected single VFLGs on a flexible carbon-cloth substrate. The insets of (**c**–**e**) show the measurement circuit diagrams of the total resistance of the VFLG and the substrate.

**Figure 7 nanomaterials-12-00971-f007:**
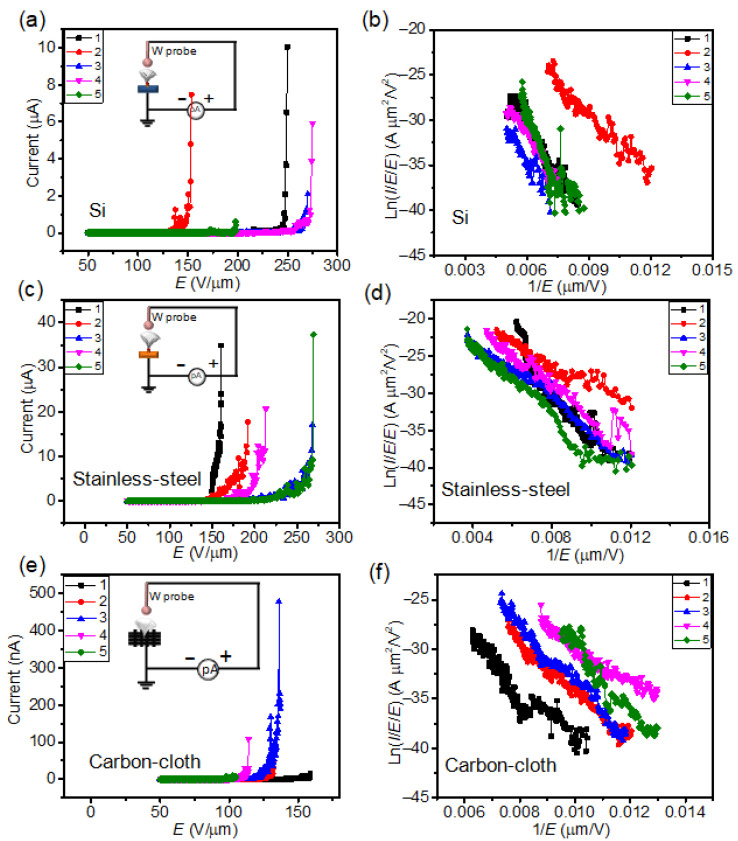
Field emission properties of single VFLGs grown on different substrates. Field emission *I*-*V* curves and *F*-*N* curves of five randomly selected single VFLGs on (**a**,**b**) Si substrates, (**c**,**d**) stainless-steel substrates, (**e**,**f**) flexible carbon-cloth substrate. Inserts are the circuits diagrams of the field emission test. All tests were carried out in DC voltage mode.

**Table 1 nanomaterials-12-00971-t001:** The vertical few-layer graphene (VFLG) growth on four substrates with the influence factors of contact angle, pretreatment and catalysis.

Substrates	Contact Angle (°)		Catalysis	After Pretreatment Growth 2 min	After Pretreatment Growth 5 min
	Before Pretreatment	After Pretreatment			
Si	66	32	No	Without VFLG	VFLG
Carbon-cloth	134	116	No	Without VFLG	Few VFLG
Quartz	50	35	No	Without VFLG	VFLG
Stainless-steel	71	107	Yes	VFLG	VFLG

**Table 2 nanomaterials-12-00971-t002:** The parameters extracted from Raman spectra for the VFLG grown on different substrates.

Substrate	*I*_D_/*I*_G_	*I*_2D_/*I*_G_	*G*_FWHM_ (cm^−1^)	2*D*_FWHM_ (cm^−1^)
Si	0.9	0.45	29	82
Stainless-steel	1.2	0.46	30	87
Carbon-cloth	0.93	0.8	39	72
Quartz	1.1	0.46	30	87

**Table 3 nanomaterials-12-00971-t003:** Comparison of the high emission current/current density/breakdown voltage of the reported single sheet/wire/tube nanomaterials.

Single Sheet/Wire/Tube Sample	VFLG	CuO Nanoneedle	SiC Nanowire	Mo Nanoscrew	LaB_6_ Nanowire	Carbon Nanotube	Multilayer Graphene
Maximum current (μA)	35	1.08	1.07	15.8	96	65	60
Current density (A/cm^2^)	7.8 × 10^5^	-	2.5 × 10^4^	2 × 10^5^	1.6 × 10^7^	-	7 × 10^7^
Breakdown voltage (V/μm)	160	9.7	220	160	320	-	-
Ref.	Our work	[35]	[36]	[37]	[38]	[39]	[40]

## Data Availability

Data presented in this article are available at request from the corresponding author.

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
