# Peer review of "Effects of Substrates on Nucleation, Growth and Electrical Property of Vertical Few-Layer Graphene"

_nanomaterials, 2022, doi:10.3390/nano12060971_

Round 1
Reviewer 1 Report
The manuscript deals the synthesis and characterization of vertical few layer graphene (VFLG) on four different substrates. The VFLG have been prepared by inductively coupled plasma enhanced chemical vapor deposition and characterized by Raman, HRTEM, electrical measurements, and field emission characterization.
Overall, the story line is clear, the authors performed a plethora of experiments to characterize their samples and the results are well-described.
In general, I suggest a careful proof-reading and maybe the use of a language-editing service. The manuscript can be accepted after some corrections. Specifically, the authors have to set their results in context to existing literature. I do not request new measurements but a careful revision and literature comparison.
Please find attached a list of comments and suggestions:
-
Ref1-6 and especially, the ref 7 and 8 which are explicitly introduced on p1 l33 to l37 should appear later again in the results section and should be compared to the results herein.
-
What is the origin of the poor adhesion on a carbon-based substrate (p3 l98 to l101)? The fact is clear and obvious but discussion is missing. One explanation is given on p4 l135/136 but this should come earlier and a reference is missing for this statement.
-
Paragraph p3 l102-104 is meaningless and can be removed.
-
Stainless steel is missing in the discussion about the surface energy. The authors nicely discuss the reduction of the contact angle after treatment for Si, carbon-cloth (CC) and quartz after Ar treatment, but the increase in contact angle on stainless-steel is not discussed (only shown in table1) Furthermore, stainless steel (SS) show similar contact angles as CC after treatment; thus, the adhesion of the VFLG on treated SS should be similar to the on CC.
-
Figure 3: If there is an iron catalyst film on CC ad on Si why is the nucleation different? It just grows on the iron? Or do the iron form particles which act as catalysts? If not why is there an influence of the substrate mediated through the iron film? References needed for discussion.
-
Figure 4 and Raman discussion: What is the peak at around 2400- 2450 cm⁻¹?
-
I am puzzling about the the resistance values given on p7 l201/202? They seem rather high for me for VFLG – can the authors compare their values to literature?
-
Doping and conductivity of Si substrate should be given in main manuscript and not only in SI.
-
Quartz is not measurable because of the high resistivity. This should be mentioned in the main manuscript.
-
The term in situ should be defined. For me in situ means that the characterization is during growth but this is not the case here.
-
Did the authors also measure the Fe-catalyst samples with respect to the field emission measurements?
-
P8 l241 “The main reason is ascribed to the large contact resistance” but the contact resistance is similar to Si and SS (100 Ohm to 5 Ohm), this should not matter too much. I agree with the low adhesion and thus eraly breakdown but not with the contact resistance. Furthermore, VFLG on Si with an higher contact resistance shows improved FE properties (as discussed in l245).
-
X-axis in figure 7 should be altered in V/µm because the discussion in main text is using the macroscopic electrical field instead of the voltage.
-
Why is the thermal conductivity an important feature of the substrate. Explanation and references should be given.
-
How were the field enhancement factors derived? Which curve was used in Fig 7? Is this a mean value?
-
How do the authors explain the significant differences of the different VFLG sheets measured on each substrate?
-
The authors need to elaborate more on their last conclusion: “The results show adhesive power and electrical properties of substrates are crucial for field emission characteristics.” I agree with the adhesion but not with the electrical properties. Why do the authors conclude that?
Author Response
February 28, 2022
Mr. Mateusz Trochowski, Dr.
Assistant Editor
Manuscript ID: nanomaterials-1611554
Type of manuscript: Article
Title: Effects of Substrates on Nucleation, Growth and Electrical Property of
Vertical Few Layer Graphene
Authors: Tianzeng Hong, Chan Guo, Yu Zhang *, Runze Zhan, Peng Zhao, Baohong
Li, Shaozhi Deng *
On behalf of all of the authors, I would like to thank you for your great efforts on evaluating our manuscript. We have carefully revised our manuscript according to the comments and suggestions of the two reviewers, and some minor grammatical mistakes, author information and descriptions in the text are also revised. The detailed response to the comments of the reviewers is included at the back of this letter. The changes have been indicated in modification trace in one of the uploaded manuscript files in the Word format.
Thank you very much for your time and consideration of our manuscript.
Sincerely,
Shaozhi Deng
*Full address of the corresponding author:
Professor Shaozhi Deng
State Key Lab of Optoelectronic Materials and Technologies,
Guangdong Province Key Laboratory of Display Material and Technology,
School of Electronics and Information Technology,
Sun Yat-sen University,
135 Xingangxi Road, Guangzhou 510275, Guangdong Province,
- R. China
Tel: +86(20)84110916
E-mail: stsdsz@mail.sysu.edu.cn
********************************************
Response to Reviewer #1
We are truly appreciated to the reviewers for the very detailed and helpful suggestions. It helped making the manuscript in good quality.
Comment:
The manuscript deals the synthesis and characterization of vertical few layer graphene (VFLG) on four different substrates. The VFLG have been prepared by inductively coupled plasma enhanced chemical vapor deposition and characterized by Raman, HRTEM, electrical measurements, and field emission characterization.
Overall, the story line is clear, the authors performed a plethora of experiments to characterize their samples and the results are well-described.
In general, I suggest a careful proof-reading and maybe the use of a language-editing service. The manuscript can be accepted after some corrections. Specifically, the authors have to set their results in context to existing literature. I do not request new measurements but a careful revision and literature comparison.
Please find attached a list of comments and suggestions:
- Ref 1-6 and especially, the ref 7 and 8 which are explicitly introduced on p1 l33 to l37 should appear later again in the results section and should be compared to the results herein.
Response:
Thanks for the reviewer’s kind advice. The difference between measurement method in these references and our result make the comparison not applicable. As we known that, the FE measurement of a single sheet and a film gets huge difference in turn on field and emission current density; the different voltage mode (DC mode and pulsing mode) would lead to different emission current density in an order of magnitude in a same sample.
In our experiment, the measurement parameter is: DC voltage mode, single FLG sheet using nano-probe anode. While, in Ref 1-6, 8, the samples are all FLG films with much large area, in Ref 7, they adopted pulse voltage mode to measure IV curve.
Based on the above starting point, we did not compare the data in the manuscript.
- What is the origin of the poor adhesion on a carbon-based substrate (p3 l98 to l101)? The fact is clear and obvious but discussion is missing. One explanation is given on p4 l135/136 but this should come earlier and a reference is missing for this statement.
Response:
Thanks for the reviewer’ kind advice. The low adhesive power of VFLG on flexible carbon-cloth substrate can be ascribed to low surface energy. Low surface energy leads to low Edes. Low Edes lead to short retention time and easy desorption of carbon radicals on substrate during nucleation process. Neither of the time and carbon radicals are sufficient for forming high crystallinity and large size nucleation interface on substrate. That is the reason for the bad adhesion of VFLG grown on flexible carbon-cloth substrate. We have added the description and discussion in the manuscript (P3 L103-104, P4 L140-145) to make the statement logical and further clear mechanism.
- Paragraph p3 l102-104 is meaningless and can be removed.
Response:
Thanks for the reviewer’ kind advice. We agree with the reviewer’ opinion and have removed the paragraph p3 L102-104.
- Stainless steel is missing in the discussion about the surface energy. The authors nicely discuss the reduction of the contact angle after treatment for Si, carbon-cloth (CC) and quartz after Ar treatment, but the increase in contact angle on stainless-steel is not discussed (only shown in table1) Furthermore, stainless steel (SS) show similar contact angles as CC after treatment; thus, the adhesion of the VFLG on treated SS should be similar to the on CC.
Response:
Thanks for the reviewer’ question. The SS truly has a similar large contact angle as CC after Ar treatment. However, due to the catalyst property of SS, the VFLG grow on SS has a good contact interface. Therefore, VFLG grow on SS still show a good adhesion. Based on this experimental result, we believed that the catalyst property of substrate can overcome the influence of surface energy. Catalyst property of substrate has a priority influence on FLG growth than surface energy. That is the reason we did not discuss the contact angle on SS in the manuscript. We have added and explained in the main manuscript P3 L129-131.
- Figure 3: If there is an iron catalyst film on CC ad on Si why is the nucleation different? It just grows on the iron? Or do the iron form particles which act as catalysts? If not why is there an influence of the substrate mediated through the iron film? References needed for discussion.
Response:
Thanks for the reviewer’ question. As we mentioned in the manuscript, iron acting as catalyst has fast VFLG nucleation rate. Therefore, an iron catalyst film on CC and Si act as the main nucleation site. During the pretreatment, the iron film firstly decomposes as iron particles, then VFLG grow on them. In the same growth condition, the VFLG only grow on iron particle, not on CC and Si due to the different VFLG nucleation rate.
- Figure 4 and Raman discussion: What is the peak at around 2400- 2450 cm⁻¹?
Response:
Thanks for the reviewer’ question. The peak at around 2400- 2450 cm⁻¹ is G* which related to disordered graphitic lattices provided by sp2-sp3 bonds. (J. Phys. Chem. C 2017, 121, 20489-20497).
- I am puzzling about the resistance values given on p7 l201/202? They seem rather high for me for VFLG – can the authors compare their values to literature?
Response:
Thanks for the reviewer’ question. Our results are similar with the reported sheet resistance of pristine graphene (0.6-1.8 KΩ/*) (Carbon, 2015, 82, 500-505.). We have compared the values to literature and added the references in the manuscript P7 L212-213.
- Doping and conductivity of Si substrate should be given in main manuscript and not only in SI.
Response:
Thanks for the reviewer’ kind advice. We have added conductivity of Si substrate in the main manuscript P8 L261-262.
- Quartz is not measurable because of the high resistivity. This should be mentioned in the main manuscript.
Response:
Thanks for the reviewer’ kind advice. In view of field emission applications the substrate should be a conductor. VFLG on quartz is not measurable because of the high resistivity of substrate. We have added the description which quartz is not measurable in the main manuscript P8 L245.
- The term in situ should be defined. For me in situ means that the characterization is during growth but this is not the case here.
Response:
Thanks for the reviewer’ kind advice. The characterization uses a nano-probe measurement system. We have revised in the manuscript.
- Did the authors also measure the Fe-catalyst samples with respect to the field emission measurements?
Response:
Thanks for the reviewer’ question. We didn’t measure the Fe-catalyst samples with respect to the field emission measurements.
- P8 l241 “The main reason is ascribed to the large contact resistance” but the contact resistance is similar to Si and SS (100 Ohm to 5 Ohm), this should not matter too much. I agree with the low adhesion and thus early breakdown but not with the contact resistance. Furthermore, VFLG on Si with a higher contact resistance shows improved FE properties (as discussed in l245).
Response:
Thanks for the reviewer’ kind advice. We agree with the reviewer’ opinion. The main reason is ascribed to the low adhesive power of the VFLG on flexible carbon-cloth. We have revised in the main manuscript P8 L253.
- X-axis in figure 7 should be altered in V/µm because the discussion in main text is using the macroscopic electrical field instead of the voltage.
Response:
Thanks for the reviewer’ kind advice. We have altered X-axis in figure 7.
- Why is the thermal conductivity an important feature of the substrate. Explanation and references should be given.
Response:
Thanks for the reviewer’ kind advice. Large amount of heat will be produced during field electron emission process. High thermal conductivity can help heat dissipation and is good for the VFLG to sustain a high current to avoid damaging cathode materials. We have added explanation and given references in the main manuscript P8 L263-264.
- How were the field enhancement factors derived? Which curve was used in Fig 7? Is this a mean value?
Response:
Thanks for the reviewer’ questions. The field enhancement factor was calculated according to F-N theory which is used for classical field electron emission process and stated as:, where A = 1.541 434 × 10−6 is the first F-N constant with the unit of A eV V−2, B = 6.830 890 is the second F-N constant with the unit of eV−3/2 V nm−1, φ is work function of the sample, and β is the field enhancement factor. Fig 7 (b), (d), (f) show F-N curves of Si, stainless-steel and carbon-cloth respectively. The results are the mean value.
- How do the authors explain the significant differences of the different VFLG sheets measured on each substrate?
Response:
Thanks for the reviewer’ question. The main reasons causing differences of the different VFLG sheets are (1) the diversity of crystal quality of each individual VFLG, (2) the diversity of contact interface. The VFLG sheets were chosen randomly during measurement, so there is an occasionality in measurement. The results showed that the field emission of VFLG will distribute over a range.
- The authors need to elaborate more on their last conclusion: “The results show adhesive power and electrical properties of substrates are crucial for field emission characteristics.” I agree with the adhesion but not with the electrical properties. Why do the authors conclude that?
Response:
Thanks for the reviewer’ question. We agree with the reviewer’ opinion. The advantage of electrical conductivity has not been proved in this manuscript. We have elaborated the conclusion in the manuscript.

Reviewer 2 Report
In this paper authors study Vertical few layer graphene (VFLG) for field emission applications.
They perform measurements using a nano probe approach in order to exclude the role of the substrate and then they study the VFLG on the substrate. They conclude that the conductivity of the VFLG sheet on the different substrates have very little differences while the contact resistance changes of a big amount. This result seems acceptable in view of the fact that the substrate has low impact on the conductivity of 2D crystal. This is a general result. See for instance Xu, Z., Ferraro, D., Zaltron, A. et al. Optical detection of the susceptibility tensor in two-dimensional crystals. Commun Phys Vol. 4, 215 (2021), where authors show that the substrate does not alter the intrinsic optical conductivity of a 2D crystal. Authors should cite this paper in support of their claim.
Substrates investigated are Si, Quartz, Stainless-steel and carbon cloth, they spam from insulating to conductive materials. In view of field emission applications I imagine the substrate should be a conductor. Why to investigate also quartz? Or Carbon cloth? This looks a bit artificial.
Author Response
February 28, 2022
Mr. Mateusz Trochowski, Dr.
Assistant Editor
Manuscript ID: nanomaterials-1611554
Type of manuscript: Article
Title: Effects of Substrates on Nucleation, Growth and Electrical Property of
Vertical Few Layer Graphene
Authors: Tianzeng Hong, Chan Guo, Yu Zhang *, Runze Zhan, Peng Zhao, Baohong
Li, Shaozhi Deng *
On behalf of all of the authors, I would like to thank you for your great efforts on evaluating our manuscript. We have carefully revised our manuscript according to the comments and suggestions of the two reviewers, and some minor grammatical mistakes, author information and descriptions in the text are also revised. The detailed response to the comments of the reviewers is included at the back of this letter. The changes have been indicated in modification trace in one of the uploaded manuscript files in the Word format.
Thank you very much for your time and consideration of our manuscript.
Sincerely,
Shaozhi Deng
*Full address of the corresponding author:
Professor Shaozhi Deng
State Key Lab of Optoelectronic Materials and Technologies,
Guangdong Province Key Laboratory of Display Material and Technology,
School of Electronics and Information Technology,
Sun Yat-sen University,
135 Xingangxi Road, Guangzhou 510275, Guangdong Province,
- R. China
Tel: +86(20)84110916
E-mail: stsdsz@mail.sysu.edu.cn
********************************************
Response to Reviewer #2
We are truly appreciated to the reviewers for the very detailed and helpful suggestions. It helped making the manuscript in good quality.
Comment:
In this paper authors study Vertical few layer graphene (VFLG) for field emission applications.
- They perform measurements using a nano probe approach in order to exclude the role of the substrate and then they study the VFLG on the substrate. They conclude that the conductivity of the VFLG sheet on the different substrates have very little differences while the contact resistance changes of a big amount. This result seems acceptable in view of the fact that the substrate has low impact on the conductivity of 2D crystal. This is a general result. See for instance Xu, Z., Ferraro, D., Zaltron, A. et al. Optical detection of the susceptibility tensor in two-dimensional crystals. Commun Phys Vol. 4, 215 (2021), where authors show that the substrate does not alter the intrinsic optical conductivity of a 2D crystal. Authors should cite this paper in support of their claim.
Response:
Thanks for the reviewer’ kind advice. We have cited that paper in the manuscript P 7 L216-217 Ref 32.
- Substrates investigated are Si, Quartz, Stainless-steel and carbon cloth, they span from insulating to conductive materials. In view of field emission applications I imagine the substrate should be a conductor. Why to investigate also quartz? Or Carbon cloth? This looks a bit artificial.
Response:
Thanks for the reviewer’ question. Besides field emission device application, VFLG has many applications such as cooling fin, supercapacitor, photoelectronic converter and so on. These potential applications require several kinds of substrates, including metal, semiconductor and insulator, also flexible substrate. That is the reason we choose these four substrates as an object of study. We want to prove that the VFLG growth method is universal and cogent. Although the result in this stage showed VFLG may not have a good performance on carbon cloth substrate, we still believe that it is worthwhile to continue to carry on more work in the future.

Round 2
Reviewer 1 Report
The manuscript deals with the synthesis and characterization of VFLG sheets on four different substrates.
The manuscript has been significantly improved during the first revision. Most of the answers and changes are appropriate.
However, I suggest that the authors comment more on the field emission results. The motivation starts with cold cathode applications, hence they have to compare their results with the existing literature. I am not satisfied with their comments regarding the introduced references 1-8, and especially 7-8. If results from this publications are highlighted, the results should be put in context and discussed with the own results. Or it should discussed in the manuscript what are the limitations.
In general, the field emission data and discussion paragraph lacks of literature comparison. There is only one single reference for the thermal conductivity given, but none for, e.g. maximum emission current, field enhancement factors or breakdown field. Discussion about turn-on field is completely missing.
Experimental details such as the evaluation of the field enhancement factor and that the values are the mean of the different flakes (btw: what is the standard deviation) have to be given in the manuscript as well.
Without a serious and detailed discussion of the field emission data, they manuscript cannot be recommended for publication.
Author Response
March 6, 2022
Mr. Mateusz Trochowski, Dr.
Assistant Editor
Manuscript ID: nanomaterials-1611554
Type of manuscript: Article
Title: Effects of Substrates on Nucleation, Growth and Electrical Property of
Vertical Few Layer Graphene
Authors: Tianzeng Hong, Chan Guo, Yu Zhang *, Runze Zhan, Peng Zhao, Baohong
Li, Shaozhi Deng *
On behalf of all of the authors, I would like to thank you for your great efforts on evaluating our manuscript. We have further revised our manuscript according to the suggestions of the reviewer. The detailed response to the comments of the reviewer is included at the back of this letter. The changes have been indicated in yellow marking in one of the uploaded manuscript files in the Word format.
Thank you very much for your time and consideration of our manuscript.
Sincerely,
Shaozhi Deng
*Full address of the corresponding author:
Professor Shaozhi Deng
State Key Lab of Optoelectronic Materials and Technologies,
Guangdong Province Key Laboratory of Display Material and Technology,
School of Electronics and Information Technology,
Sun Yat-sen University,
135 Xingangxi Road, Guangzhou 510275, Guangdong Province,
- R. China
Tel: +86(20)84110916
E-mail: stsdsz@mail.sysu.edu.cn
********************************************
Response to Reviewer #1
We are truly appreciated to the reviewer for the very detailed and helpful suggestions. Those comments are very helpful for revising and improving our paper.
Comments
The manuscript deals with the synthesis and characterization of VFLG sheets on four different substrates.
The manuscript has been significantly improved during the first revision. Most of the answers and changes are appropriate.
However, I suggest that the authors comment more on the field emission results. The motivation starts with cold cathode applications, hence they have to compare their results with the existing literature. I am not satisfied with their comments regarding the introduced references 1-8, and especially 7-8. If results from this publications are highlighted, the results should be put in context and discussed with the own results. Or it should discussed in the manuscript what are the limitations.
In general, the field emission data and discussion paragraph lacks of literature comparison. There is only one single reference for the thermal conductivity given, but none for, e.g. maximum emission current, field enhancement factors or breakdown field. Discussion about turn-on field is completely missing.
Experimental details such as the evaluation of the field enhancement factor and that the values are the mean of the different flakes (btw: what is the standard deviation) have to be given in the manuscript as well.
Without a serious and detailed discussion of the field emission data, they manuscript cannot be recommended for publication.
Response:
Thanks for the reviewer’s kind advice. According to the suggestions, we have added the description for the discussion of the field emission characteristics (P8 L251-256, P9 L285-293 and P11 L300-302) and the evaluation of the field enhancement factor (P9 L274-281), which the literature is also supplemented accordingly (P14 L432-448).
Revision: As shown in the following yellow markings, corresponding in the main manuscript P8 L251-256, P9 L274-281, P9 L285-293, P11 L300-302 and P14 L432-448.
The field electron emission characterization of VFLG on Si, stainless-steel and flexible carbon-cloth substrates were measured to further explore the influence of substrates on VFLG. VFLG on quartz is not measurable because of the high resistivity of substrate. Field emission characterization of a single VFLG sheet was characterized using the same nano-probe measurement system. The voltage applied to anode namely the tungsten probe. The substrate was set as ground to be cathode emitter. The distance of the VFLG and probe was 1 μm. The inset of Figure 7 shows the circuit diagram of VFLG field electron emission. It is obviously that the field emission characterization of the VFLG is influenced by the substrate. The turn-on field of VFLG on Si, stainless-steel and flexible carbon-cloth substrate are 158±30 V/μm, 118±16 V/μm and 114±16 V/μm respectively at a current 1 nA. At the beginning of field emission, temperature and adhesive power have not influence on field emission. Turn-on field of VFLG on stainless-steel and flexible carbon-cloth substrate is similar. Schottky contact of VFLG on Si substrate leads to large contact resistance resulting high turn-on field. The VFLG grown on flexible carbon-cloth substrate shows the poorest emission current which is only 0.5 μA under the breakdown voltage, correspondingly, the current density is 1.4×104 A/cm2 (Figure 7 (e)). The main reason is ascribed to the low adhesive power of the VFLG on flexible carbon-cloth, where the VFLG easily detach from flexible carbon-cloth substrate causing vacuum breakdown. The VFLG grown on Si and stainless-steel substrates presents better field emission current. The maximum emission current of VFLG grown on Si substrate reached 10 μA at 250 V/μm before vacuum breakdown, correspondingly, the current density is 2.2×105 A/cm2 (Figure 7 (a)). The maximum emission current of VFLG grown on stainless-steel substrate reached 35 μA at 160 V/μm before vacuum breakdown, correspondingly, the current density is 7.8×105 A/cm2 (Figure 7 (c)). The difference of the two is attributed to electrical and thermal conductivity of substrates. The electrical conductivity of n-Si and stainless-steel is 1.56×10-3 S/m, 9×106 S/m and the thermal conductivity of n-Si and stainless-steel is 133 W/m K, 14 W/m K (supporting information table S1). High thermal conductivity can help heat dissipation and is good for the VFLG to sustain a high current during field electron emission process [7]. Excellent electrical conductivity and thermal conductivity strongly promotes the field emission characterization of the VFLG. The corresponding F-N plot is linear in low field regions and fit with field emission F-N theory (Figure 7 (b) (d) (f)). The field enhancement factor of VFLG can be calculated according to F-N theory, which is applied for classical field emission process and described as: (A=1.541434×10-6 A eV V-2, B=6.830890 eV V nm-1, β field enhancement factor, φ work function of sample, E electrical field). The work function of VFLG can be assumed to be constant at 4.5 eV [33]. The field enhancement factor of the VFLG on Si, stainless-steel and flexible carbon-cloth substrate is 21±2.4, 33±9.1 and 26±5.7 respectively. The small field enhancement factor can be ascribed to short distance of probe and VFLG [34]. Similar field enhancement factor of VFLG further indicate important of substrate for field emission of VFLG. The results show adhesive power and electrical properties of substrates are crucial for field emission characteristics of VFLG.
Finally, the maximum field emission current of the VFLG is compared with some single sheet/wire/tube nanomaterials with excellent field emission performance in Table 3. The VFLG is better than CuO nanoneedle, SiC nanowire and Mo nanoscrew, but not good as LaB6 nanowire, carbon nanotube and multilayer graphene. Current density of VFLG is larger than SiC nanowire and Mo nanoscrew, but lower than LaB6 nanowire and multilayer graphene. Breakdown voltage of VFLG is lower than SiC nanowire and LaB6 nanowire. The limitations of the VFLG field emission characteristic can be ascribed to low adhesive power and low crystallinity. Improving the quality of VFLG is crucial for excellent field emission characteristics.
Table 3. Comparison of the high emission current/current density/breakdown voltage of the reported single sheet/wire/tube nanomaterials.
|
Single sheet/wire/tube sample |
VFLG |
CuO nanoneedle |
SiC nanowire |
Mo nanoscrew |
LaB6 nanowire |
Carbon nanotube |
Multilayer graphene |
|
Maximum current (μA) |
35 |
1.08 |
1.07 |
15.8 |
96 |
65 |
60 |
|
Current density (A/cm2) |
7.8×105 |
- |
2.5×104 |
2×105 |
1.6×107 |
- |
7×107 |
|
Breakdown voltage (V/μm) |
160 |
9.7 |
220 |
160 |
320 |
- |
- |
|
Ref. |
Our work |
[35] |
[36] |
[37] |
[38] |
[39] |
[40] |
- Gugel, D. Niesner, C. Eickhoff, S. Wagner, M. Weinelt, T. Fauster. Two-Photon Photoemission from Image-Potential States of Epitaxial Graphene. 2D Mater., 2015, 2, 045001.
- Q. Wang, M. Wang, Z.H. Li, Y.B. Xu, P. M. He. Modeling and Calculation of Field Emission Enhancement Factor for Carbon Nanotubes Array. Ultramicroscopy, 2005, 102, 181-187.
- L. Liu, L. Zhong, Z. Y. Peng, Y. B. Song, W. Chen. Field Emission Properties of One-Dimensional Single CuO Nanoneedle by in situ Microscopy. J. Mater. Sci., 2010, 45, 3791-3796.
- Zhao , Y. Zhang, S. Tang, R. Z. Zhan, J. C. She, J. Chen, N. S. Xu, S. Z. Deng. Effect of Piezoresistive Behavior on Electron Emission from Individual Silicon Carbide Nanowire. Nanomaterials, 2019, 9, 981.
- Shen, N. S. Xu, S. Z. Deng, Y. Zhang, F. Liu, J. Chen. A Mo Nanoscrew Formed by Crystalline Mo Grains with High Conductivity and Excellent Field Emission Properties. Nanoscale, 2014, 6, 4659-4668.
- B. Gan, L. X. Peng, X. Yang, Y. Tian, N. S. Xu, J. Chen, F. Liu, S. Z. Deng. A Moderate Synthesis Route of 5.6 mA-current LaB6 Nanowire Film with Recoverable Emission Performance Towards Cold Cathode Electron Source Applications. RSC Adv., 2017, 7, 24848.
- S. Wang, L. M. Peng, J. Y. Wang, Q. Chen. Electron Field Emission Characteristics and Field Evaporation of a Single Carbon Nanotube. J. Phys. Chem. B., 2005, 109, 110-113.
- Nakakubo, K. Asaka, H. Nakahara, Y. Saito. Evolution of Field Electron Emission Pattern from Multilayered Graphene Induced by Structural Change of Edge. Appl. Phys. Express, 2012, 5, 055101.

Round 3
Reviewer 1 Report
During the 2nd revision the authors further improved the manuscript especially with respect to the discussion of the field emission characteristics.
The publication can now be recommended for publication in MDPI Nanomaterials.